# Clinical Nurses’ Intention to Use Defibrillators in South Korea: A Path Analysis

**DOI:** 10.3390/healthcare11010061

**Published:** 2022-12-26

**Authors:** Dongchoon Uhm, Gye-Hyun Jung

**Affiliations:** 1Department of Emergency Medical Technology, Daejeon University, 62 Daehak-ro, Dong-gu, Daejeon 300-716, Republic of Korea; 2Department of Nursing, Jeonbuk Science College, Jeonbuk 56204, Republic of Korea

**Keywords:** behavior intention, clinical nurses, defibrillators, image, organizational support

## Abstract

This study aimed to investigate factors affecting behavioral intentions to use defibrillators among clinical nurses in South Korea, using a modified predictive technology acceptance model 2 (TAM 2) that facilitates understanding of what prompts apparent spontaneous actions. This descriptive, cross-sectional study included 467 clinical nurses with more than 6 months of clinical experience. The path analysis results showed that the variables directly affecting the intention to use a defibrillator among clinical nurses were the image, organizational support, perceived ease of use, and perceived usefulness. Moreover, job autonomy and total career indirectly affected the intention to use a defibrillator. Clinical nurses need to know how to use defibrillators and be able to act promptly on patients with in-hospital cardiac arrest (IHCA). Organizational managers need to improve the work environments of clinical nurses accordingly. Additionally, it is necessary to establish a policy foundation to ensure the development of competence and job autonomy that can lead to the use of defibrillators by improving various factors, including anxiety or reluctance of nurses to use defibrillators in acute cardiac arrest.

## 1. Introduction

Lately, there has been much interest and development in nurses’ resuscitation performance as cardiopulmonary resuscitation (CPR) team members [1]. Although there is no promise for the extent of nurses’ domain with cardiac arrest (CA) in the Korean medical system, it cannot be doubted that nurses should be able to perform CPR with defibrillation regardless of the time and place of CA.

According to evaluation criteria for healthcare accreditation in South Korea, medical staff (doctors, nurses) and technicians who have direct contact with patients should receive CPR training once every 2 years [2]. There is no legal problem with clinical nurses using a defibrillator when it is required [3]. However, trained clinical nurses are hesitant to use a defibrillator and feel it is a challenge despite the actual situation that requires defibrillation. In previous studies [4,5,6], a lack of knowledge of shockable rhythms and defibrillation, insecurity and anxiety about using defibrillators, recognizing it as a doctor’s assistant job, or fear of negative outcomes were reported as the reasons clinical nurses do not actively use the defibrillator. In particular, Korean healthcare is characterized by a rather shy culture and hierarchical structure of doctors and nurses [7]. As a result, even if warning signs of CA are recognized and reported in a timely manner, the patient may not be appropriately warned, depending on the individual’s experience, attitude, working environment, and location of nurses or doctors. It is important that doctors and nurses work as a team; nurses can respond first, and doctors should be able to cooperate when necessary [8]. The incidence of CA in South Korea has gradually increased from 21,905 cases in 2008 (44.3 per 100,000 population) to 30,539 cases in 2018 (59.5 per 100,000 population) [9]. Considering the rapidly aging society in South Korea, the incidence of CA is likely to increase further. In-hospital cardiac arrest (IHCA) patients with specific cardiac diseases, monitored CA or witnessed CA were related to initial shockable rhythm, and within 3 minutes of a patient who collapsed receiving defibrillation, the better their chances of survival [10]. A defibrillation is a powerful tool. However, it is important to know when to use defibrillation to reset the abnormal rhythm and when not to. 

The Korean Association of Cardiopulmonary Resuscitation (KACPR) developed the Korean advanced life support-provider (KALS-P) guidelines in 2012 so that healthcare providers working on the front line can undertake professional resuscitation IHCA. The guidelines consist of the two pathways for CA. There are two shockable rhythms, such as ventricular fibrillation and pulseless ventricular tachycardia, and two non-shockable rhythms, including asystole and pulseless electrical activity [11]. The Korean Nurses Association has also recognized and implemented this program as job training for nurses since 2015 [12]. 

Today, more than ever, the active behavior of clinical nurses is required in defibrillator use. However, previous studies have reported only the reasons why they are reluctant to participate in using a defibrillator. Now, research is needed to predict the active behavior of clinical nurses to use a defibrillator.

Therefore, this study aims to predict the behavioral intention of clinical nurses to use a defibrillator by applying the technology acceptance model 2 (TAM 2) theory, which explains how users come to accept and use a technology.

The technology acceptance model (TAM), developed by Davis [9] and based on reasoned action theory, has been used to explain a potential user’s behavioral intention to make use of technological innovation [10] and employed as a useful predictive model to explain what makes users act voluntarily. The most important concepts underlying this model are perceived usefulness and ease of use [9]. Perceived usefulness has been defined as the degree to which a user believes that using a particular system or technology will improve their performance. Perceived ease of use has been described as the degree to which a user believes that using a particular system or technology will require little or no effort [8].

Ongoing research has been conducted to refine the limitations of TAM. Venkatesh and Davis [10] upgraded the TAM to create TAM 2 by including additional key determinants, including organizational intervention, to better understand the determinants of perceived usefulness. The TAM 2 model aims to explain perceived usefulness and usage intentions in terms of social influence (subjective norm, voluntariness, and image) and cognitive instrumental processes (job relevance, output quality, result demonstrability, and perceived ease of use) [10].

TAM consistently predicts a good portion of the variation in the intention of healthcare professionals to accept new technology, with the reported percentage of variance explained in the dependent variable being reasonably high in a clinical context [11]. Researchers have applied the extended TMA 2 to health information systems to understand the behavioral intentions of users by applying external variables related to the clinical context [11,12,13]. The system users’ behavioral intentions may be determined by the personal experience of users, subjective norms, and an image of social influence [12,14]. Regarding organization factors, organizational support correlated [15] and job-related variables of nurses influenced the intention to use technology [13]. The perceived ease of use positively influences perceived usefulness and the intention of nurses to accept to use of information technology [11,13,15].

Therefore, this study aimed to explain the perceived usefulness and usage intentions of clinical nurses to use defibrillators in terms of social variables (image, subject norm), organizational variables of job relevance (job autonomy, organizational support) as well as user’s personal characteristics (total career) and perceived ease of use, using a conceptual framework based on the TAM 2 (Figure 1).

## 2. Materials and Methods

### 2.1. Research Design

A descriptive, cross-sectional design was used to administer a self-report questionnaire to investigate the factors affecting the behavioral intention to use defibrillators among clinical nurses in South Korea, based on the TAM 2. The questionnaire took approximately 10–15 minutes to complete.

### 2.2. Sample and Study Context

Using a convenience sampling method, participants were recruited from three university hospitals and five general hospitals in a large city in South Korea. The inclusion criterion was clinical nurses with more than six months of experience. Part-time nurses and nurses with fewer than six months of experience were excluded. The principal researcher contacted the directors of eight hospitals to obtain permission for recruitment, and data were collected between May and August 2019.

This study was conducted to construct the structural model by path analysis with eight observational variables. For maximum likelihood estimate, the sample should be at least 200; the more complicated the path, the larger the sample size so that the tolerance error can be reduced [16,17]. Therefore, considering the dropout rate, we recruited 500 clinical nurses to achieve a reasonable sample for this study. A total of 500 questionnaire packages with cover letters were distributed with a return envelope addressed to the researcher, and 462 questionnaires were analyzed (response rate = 92.4%).

### 2.3. Measurements

The questionnaire consisted of six sections as follows: demographic characteristics (gender, educational level, total career, participation in CPR, experience of chest compression during CPR, experience of defibrillation in CPR, training in defibrillation use, basic life support-health care professionals [BLS-HCP] certificate, and advanced cardiovascular life support-provider [ACLS-P] certificate), social variables (image and subjective norm), organizational variables (job autonomy and organizational support), perceived ease of use of defibrillators, perceived usefulness of defibrillators, and intention to use defibrillators, and comprised 43 questions in all. All study variables apart from the demographic variables were assessed using a five-point scale, with answers ranging from (1) “strongly disagree” to (5) “strongly agree.”

Confirmatory factor analysis was applied to the study variables to assess its construct validity. The questionnaire was translated into Korean and back-translated into English to check the validity of the translation, and its content validity was assessed by three experts (one nursing director and two nursing professors). In addition, a pilot test (*n* = 50) was performed prior to conducting the study.

#### 2.3.1. Intention to Use Defibrillators

This measurement was developed by Venkatesh and Davis [10] and modified to identify the intention to use defibrillators for CA in this study. It consists of four items, with a higher score indicating a higher level of motivation or willingness to use defibrillators. The Cronbach’s alpha was 0.82 in a study by Venkatesh and Davis [10] and 0.92 in this study.

#### 2.3.2. Perceived Usefulness

This measurement was developed by Venkatesh and Davis [10]. It consisted of four items with a higher score indicating a higher level of perceived usefulness of defibrillators in CA. The Cronbach’s alpha in a study by Venkatesh and Davis [10] was 0.87, and 0.92 in this study.

#### 2.3.3. Perceived Ease of Use

This measurement was developed by Venkatesh and Davis [10], and it consisted of four items with a higher score indicating that clinical nurses perceived greater ease of use of defibrillators in CA. The Cronbach’s alpha in a study by Venkatesh and Davis [10] was 0.86, and 0.90 in this study.

#### 2.3.4. Social Variables

##### Image

This measurement was developed by Baek et al. [18] and comprised four items. The higher the score, the more nurses perceived that their status was enhanced in their social systems by using defibrillators in CA. The Cronbach’s alpha in the study by Baek et al. [18] was 0.86, and 0.82 in this study. 

##### Subjective Norm

This measurement was developed by Venkatesh and Davis [10], and it comprised four items. The higher the score, the higher the clinical nurses’ perception was that most people important to them thought clinical nurses should use defibrillators for CA. The Cronbach’s alpha in a study by Venkatesh and Davis [10] was 0.81, and 0.89 in this study. 

#### 2.3.5. Organizational Variables

##### Job Autonomy

This was measured using the Korea Occupational Stress Scale by Chang et al. [19]. It consisted of seven items. The higher score indicated a higher degree of clinical nurse job autonomy for the use of defibrillators in CA. The Cronbach’s alpha in the study by Chang et al. [19] was 0.61 and 0.79 in this study.

##### Organizational Support

This measurement has been refined by a study by Lee [20] and consisted of six items. The higher the score, the higher the clinical nurses’ awareness of organizational support for the use of defibrillators in CA. The Cronbach’s alpha in a study by Lee [20] was 0.72 and 0.75 in this study.

### 2.4. Data Analysis

Data were analyzed using IBM SPSS Statistics 22 and IBM SPSS AMOS 22 (IBM Corp., Armonk, NY, USA). Statistical analyses included descriptive statistics, Pearson’s correlation coefficients, and path analysis. The normal distribution of the study variables was assessed using skewness and kurtosis. Multicollinearity was assessed using tolerance limits and variation inflation factor (VIF) values. Path analysis was performed to build the structural model of the study. To confirm the goodness-of-fit of the hypothetical and modified models in this study, we determined the absolute fit indexes (χ^2^ and χ^2^/df < 3), the goodness-of-fit index (GFI > 0.90), the adjusted goodness-of-fit index (AGFI > 0.90), the comparative-fit index (CFI > 0.90), the Tucker–Lewis index (TLI > 0.90), and the root mean squared error of approximation (RMSEA < 0.05) [16]. Path coefficient estimation and effect analysis were performed using the maximum-likelihood method. The bootstrapping method was used to verify the significance of the direct and indirect effects and the total effect.

### 2.5. Ethical Considerations

This study was conducted in accordance with the principles of the Declaration of Helsinki. This study was approved by the institutional review board (no. 1040647-202004-HR-013-01). Participation was voluntary and anonymous. Prior to the study, we received consent from all participants, who were informed that consent could be withdrawn at any time during this study without consequences.

## 3. Results

### 3.1. Demographic Characteristics

Most participants were female (97.0%), and 54.5% of the nurses had clinical experiences of less than five years. Most of the work departments of participants were general units (Medical/Surgical, OBGY/PED) (62.9%), followed by special units (28.1%). Additionally, 74.0% of the participants had participated in CPR, and 45% had experienced administering chest compression. However, only 11.7% had the experience of using defibrillation. Most participants (94.8%) had been trained in the use of defibrillators, and 63.4% had a BLS-HCP certificate (Table 1).

### 3.2. Mean Scores, Correlation among the Study Variables

The mean scores for image, subjective norm, job autonomy, and organizational support were 3.53 (0.72), 2.71 (0.82), 3.12 (0.59), and 3.19 (0.59), respectively. The mean scores for perceived ease of use of defibrillators, perceived usefulness of defibrillators, and intention to use defibrillators were 33.3 (0.73), 4.02 (0.67), and 3.54 (0.71), respectively (Table 2).

There was a significant correlation between all variables except for the total career, image, organizational support, and intention to use a defibrillator. The measured variables were found to be the normality assumption, as standardized skewness values ranged from 0.01 to 1.07, and standardized kurtosis values ranged from −0.15 to 0.93. The multivariate kurtosis value did not meet mardia’s criterion and did not meet for multivariate normality. When multivariate normality is not satisfied, a problem of upward deflection of the threshold value may occur in parameter estimation [16]. However, the data of this study can be reliably estimated parameters using maximum likelihood method even if univariate normality is met and multivariate normality assumption is not met [21]. Therefore, the path analysis was conducted by applying the maximum likelihood estimation method. To determine multicollinearity, we confirmed that the correlation coefficient remained below 0.60. Additionally, the tolerance limit exceeded 0.1 (0.69–0.92), and the VIF was 1.08–1.45, which was less than 10, indicating the absence of multicollinearity.

### 3.3. Fitness of the Hypothetical Model

The model fitness of the hypothetical model was as follows: χ^2^ = 213.54 (*p* < 0.005) and χ^2^/df = 21.35, GFI = 0.91, AGFI = 0.90, CFI = 0.86, TLI = 0.74, and RMSEA = 0.21. Among fit indexes, GFI, AGFI, and CFI were good. However, the significance probability of χ^2^ was less than 0.05, χ^2^/df exceeded 3.0, and RMSEA exceeded 0.08. In addition, six of the total 18 paths were not statistically significant, requiring model modification.

### 3.4. Model Modification

The existing variables were maintained and modified stepwise using a critical ratio and modification index in consideration of the logical validity and theoretical background of the model. For simplicity, the optimal model was identified by modifying each path (Figure 2).

First, two paths were deleted using a fixed index, the path between image and perceived ease of use of a defibrillator, which was not statistically significant, and that between subjective norm and perceived ease of use of a defibrillator were deleted. Although the significance probability of χ^2^ was less than 0.05, χ^2^/df exceeded 3.0. Second, five paths were added by referring to logical validity, theoretical background, and modification index. We added the paths between total career and intention to use a defibrillator, image and intention to use a defibrillator, and organizational support and intention to use a defibrillator. Although the paths between subjective norm and intention to use a defibrillator and that between job autonomy and intention to use a defibrillator were not statistically significant, it was decided not to delete it from the model after comparing the logical validity and total career.

The modified model had the following values: χ^2^ = 10.41 (*p* = 0.005) and χ^2^/df = 5.209, GFI = 0.99, AGFI = 0.92, CFI = 0.99, TLI = 0.86, and RMSEA = 0.09. The *p*-values of χ^2^ and χ^2^/df were affected by the sample size and complexity of the model, but GFI, AGFI, CFI, and TLI were close to 1, and RMSEA was close to 0.08 [18]; therefore, the fit of the modified model was good.

### 3.5. Path Coefficient and Effects of the Modified Model

The final path analysis results showed that 13 of the 16 paths in total were significantly affected (Figure 2). The direct, indirect, and total effects of the research variables are listed in Table 3.

Regarding the total effect on the intention to use a defibrillator was influenced by perceived ease of use of a defibrillator (*β* = 0.50, 99% CI [0.40, 0.57], *p* = 0.001), organizational support (*β* = 0.32, 99% CI [0.23, 0.41], *p* = 0.001), perceived usefulness of a defibrillator (*β* = 0.23, 99% CI [0.13, 0.32], *p* = 0.001), image (*β* = 0.22, 99% CI [0.14, 0.31], *p* = 0.001), job autonomy (*β* = 0.10, 99% CI [0.04, 0.24], *p* = 0.004), and total career (*β* = 0.08, 99% CI [0.01, 0.17], *p* = 0.027), in order. Meanwhile, the subjective norm showed significant indirect effects on intention to use a defibrillator (*β* = −0.03, 99% CI [−0.06, −0.01], *p* = 0.001); however, the total effect was not significant (*β* = 0.02, 99% CI [−0.03, 0.10], *p* = 0.364). The explanatory power was 51%.

Concerning the total effect on the mediating variables, the perceived ease of use of a defibrillator was influenced by total career (*β* = 0.34, 99% CI [0.26, 0.41], *p* = 0.001), organizational support (*β* = 0.28, 99% CI [0.19, 0.38], *p* = 0.001), and job autonomy (*β* = 0.16, 99% CI [0.06, 0.26], *p* = 0.001), and the explanatory power was 25%. The perceived usefulness of a defibrillator was influenced by image (*β* = 0.34, 99% CI [0.23, 0.43], *p* = 0.001), the perceived ease of use of a defibrillator (*β* = 0.26, 99% CI [0.15, 0.35], *p* = 0.001), organizational support (*β* = 0.26, 99% CI [0.17, 0.34], *p* = 0.001), the subjective norm (*β* = −0.15, 99% CI [−0.24, −0.06], *p* = 0.001), and total career (*β* = 0.19, 99% CI [0.11, 0.27], *p* = 0.001), but job autonomy was not significant (*β* = 0.05, 99% CI [−0.04, 0.14], *p* = 0.294). The explanatory power was 30%.

## 4. Discussion

This study aimed to develop a path model to identify the factors affecting the intention to use defibrillators among clinical nurses in three university hospitals and five general hospitals in a large city in South Korea based on TAM 2 [1,10,11].

The result of path analysis in this study revealed that the intention to use defibrillators among clinical nurses was directly affected by total career, image, organizational support, the perceived ease of use of defibrillators, and the perceived usefulness of defibrillators, and indirectly affected by job autonomy.

In demographic variables, total career (i.e., clinical experience) directly affected the perceived usefulness, perceived ease of use of defibrillators, and intention to use defibrillators. These results were similar to those of a previous study [22]. While repeated training of BLS conducted annually to clinical nurses is also beneficial, the lack of skilled and experienced nurses is one of the successful resuscitation barriers [23]. Therefore, the relevant institutions must be able to take into account not only BLS education but also skilled nurses and teams, adopt simulation education methods for the various CA situations in the hospital, and allow clinical nurses to actively consider the use of defibrillators [24,25,26].

In Social variables, the image directly affected perceived usefulness and the intention to use defibrillators. It can be interpreted that the use of defibrillators by clinical nurses is not only an opportunity to show the competencies of nurses to all medical personnel who participated in CA but also an opportunity to improve their image. These results were consistent with the results reported by Bennani and Oumlil [14]. Since the image has a positive impact on performance [27], CPR training for clinical nurses should create professional images, [28] and improve knowledge and competence related to CPR and defibrillation. To this end, the curriculum for CPR and defibrillation should be developed from the undergraduate course, and the nursing students should be educated by including clear policies and procedures along with continuous updates [24,26].

The subjective norm variable directly and negatively affected the perceived usefulness of defibrillators but not the intention to use defibrillators which was consistent with the result reported by Andrews et al. [29]. The use of defibrillators may be interpreted as negative by clinical nurses themselves and surrounding medical personnel, including co-workers. The social norms in a hospital environment not only negatively affect the knowledge and values of nurses but also could be an obstacle to first aid decisions in an emergency situation such as CA [30]. There is a necessity to expand awareness that defibrillation is not a first aid that only doctors can perform and that clinical nurses also have the ability to do it; hospital authorities need to support the use of defibrillators by clinical nurses and consensus among medical personnel is required to do that.

In the organizational variables, job autonomy directly affected the perceived ease of use of defibrillators but not the perceived usefulness of defibrillators and indirectly affected the intention to use defibrillators through fully mediated effects. These results were consistent with the results that job-related variables affected the intention to use technology, reported by Nadri et al. [13]. Job autonomy has been related to individual knowledge and judgment, the capacity to interact with other colleagues [31], and also positively associated with safety performance [4]. However, one reason clinical nurses are reluctant to use defibrillators in South Korea is that the scope of work is specified as that undertaken by a medical assistant in Korean medical legislation [4]. Although using defibrillators among clinical nurses is likely to improve patient survival rates, if nurses are held responsible for negative outcomes, less defibrillator use can be expected. Therefore, organizational support for defibrillator use is needed, and clinical nurses need to maximize their job autonomy in relation to CA within the permitted scope of work.

The organizational support directly affected the perceived ease of use and perceived usefulness of defibrillators but indirectly affected the intention to use defibrillators through a partially mediated effect. It may be interpreted that systematic support of hospital authorities for clinical nurses’ use of defibrillators can make clinical nurses aware of the perceived ease of use and perceived usefulness of defibrillators and further enhance their intention to use defibrillators. These results were consistent with those of Hui et al. [13]. Organizational support, such as changes in organizational policies, needs to facilitate clinical nurses’ intention to use defibrillators in response to in-hospital cardiac arrest (IHCA) [28]. Additionally, to promote clinical nurse-led use of defibrillators successfully, clinical nurses need up-to-date knowledge and skills concerning CPR and be familiar with the relevant protocols [15]. To this end, the organizations concerned should not only provide on-site education and training to clinical nurses with medical personnel in simulation situations but also conduct feedback and debriefing for evaluations associated with CA treatment [1,3,22]. As Kaplow and Mota [32] have suggested, debriefing helps highlight the efforts and better parts of clinical nurses and identify opportunities for improvement. Moreover, addressing emotions due to stressful events such as cardiopulmonary arrest must be done in the debriefing process, and all participants must have the opportunity to comment on the overall evaluation process and give equal voices. Furthermore, continuous and repetitive education with medical personnel in simulation situations supports job autonomy, which is likely to lead to greater safety in the care and a deepening of knowledge among professionals, as well as providing professional satisfaction and improving patient outcomes [1,3].

Finally, we found that the standardized total effect of the perceived ease of use on the intention to use defibrillators was higher than the perceived usefulness. This result suggests that clinical nurses play to use defibrillators a more important role in the perceived ease of use than the perceived usefulness of defibrillators, which was consistent with the result reported by Nadri et al. [1]. According to a study by Strrk et al. [26], nurses reported restrictions on work for AEDs use and a lack of knowledge and contextual training for AEDs use. Since IHCAs can occur without AED, it should be a work regulation so that nurses can perform efficient work as first responders [26]. To this end, team-based simulation training should be conducted in accordance with the field conditions to consider the proximity of defibrillator deployment and to communicate smoothly with teamwork. Clinical nurses should be actively and repeatedly trained using defibrillators used in the ward to familiarize themselves with the use of defibrillators and have a clearer understanding of how their use of defibrillators will benefit them and their patients [1]. In addition, Efforts should also be made to actively support related organizations and prepare guidelines.

## 5. Implementation

In IHCA situations, most clinical nurses are the first to be discovered, and their responses can affect the survival rate of cardiac arrest patients [24]. However, skilled clinical nurses are also hesitant to use defibrillators. To solve this problem, first of all, repeated annual BLS training for clinical nurses is also beneficial, but novice nurses must be able to team up with skilled nurses to learn how to use defibrillators and perform IHCAs quickly. In addition, institutions need to consider the installation of AED in each ward for safe and rapid CPR with defibrillation. Second, educators should develop a curriculum and operate it practically so that nursing students can learn a variety of cardiopulmonary resuscitation techniques from undergraduate courses and increase their ability and confidence. Through this training, clinical nurses can improve their own image. Third, to improve survival rate in IHCA situations, training such as KALS-P (Korean Advanced Life Support-Provider)/ACLS-P, as well as BLS, should be carried out continuously and repeatedly to ensure that clinical nurses acquire up-to-date knowledge and skills in CPR. To this end, organizational support and communication among medical personnel, such as changes in organizational policy, will be essential. Finally, support and policy changes from relevant institutions are required to ensure competency development and job autonomy in relation to the use of defibrillators within the permitted work scope of nurses to enhance patient outcomes and clinical nurse professional satisfaction.

## 6. Strengths and Limitations

The strength of this study was that this study assessed factors affecting the intention to use defibrillators among clinical nurses in South Korea through setting variables based on TAM 2. The results of this study provided an opportunity to strengthen TAM 2 capabilities as a useful theoretical tool in the healthcare context. This study had several limitations. First, the results of the study should be interpreted with caution because the path model could not include other confounding variables, such as doctors and policies of the South Korean medical system that may be associated with clinical nurses’ intention to use defibrillators. Furthermore, one-quarter of the nurses in this study were not exposed to CPR, so this cannot be ruled out as a limitation to their approach to in-hospital CPR situations. In future, it will be necessary to identify the factors affecting the use and non-use of defibrillators through a longitudinal study including only nurses exposed to CPR situations. Second, this study’s sample was not fully representative of clinical nurses in South Korea, and Third, although the pilot test was conducted to minimize subjective and reporting bias on the self-reported questionnaire used in this study, the social desirability bias and acquiescence responses bias could not be excluded [33]. Therefore, the results of this study might not be generalizable to all clinical nurses in Korea.

## 7. Conclusions

The use of defibrillators by clinical nurses IHCA can be an opportunity to demonstrate their abilities as well as improve their image to other healthcare professionals. Therefore, clinical nurses need to know how to use defibrillators and be able to act promptly IHCA. Organizational managers need to improve clinical nurses’ work environments accordingly.

Even if clinical nurses are given opportunities for ongoing education and training, they cannot improve their abilities without actively using defibrillators. They need to engage more directly in using defibrillators, with the support of organizational managers, to enhance their willingness to use defibrillators. Such active engagement among clinical nurses is likely to lead to positive effects in terms of professional satisfaction and patient outcomes. In addition, according to the needs of the times, it is necessary to establish a policy foundation to ensure the development of competence and job autonomy that can lead to the use of defibrillators by improving various factors that are anxious or reluctant to use defibrillators in acute CA.

## Figures and Tables

**Figure 1 healthcare-11-00061-f001:**
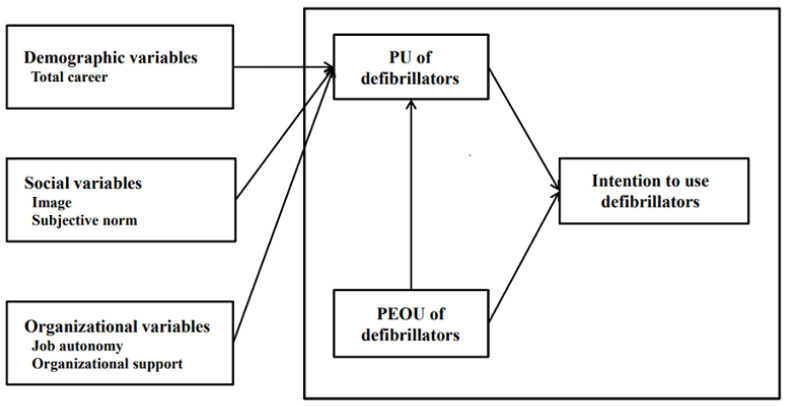
Research framework on intention to use defibrillators of clinical nurses based on TM2 model.

**Figure 2 healthcare-11-00061-f002:**
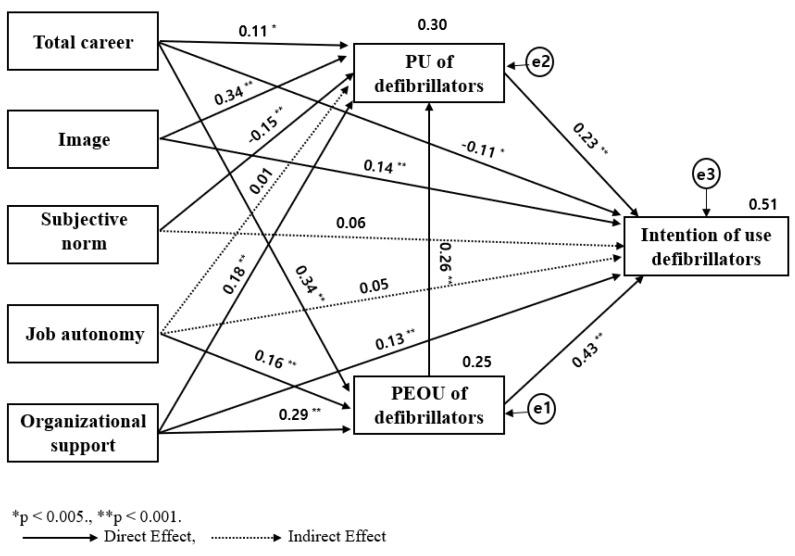
Modified model of factors influencing intention to use defibrillators of clinical nurses.

**Table 1 healthcare-11-00061-t001:** General Characteristics of Clinical Nurses (*N* = 462).

Characteristics	Category	*N*	%
Gender	Male	14	3.03
	Female	448	96.96
Education level	Diploma	168	36.36
	Bachelor	261	56.49
	≥Master	33	7.14
Total career (Years)	<5	252	54.54
	5–9	111	24.02
	10–14	52	11.25
	≥15	47	10.17
The work department of nurses (unit)	Medical/Surgical	269	58.2
	OBGY/PED	22	4.7
	Special (ICU/ED/OR)	130	28.1
	Others	41	8.9
Participation of CPR	yes	342	74.02
	no	120	25.97
Experience chest compression in CPR	0	254	54.97
	1–2	135	29.22
	≥3	73	15.80
Experience of defibrillation in CPR	0	408	88.31
	1	32	6.92
	2	13	2.81
	≥3	9	1.94
Education of defibrillation	0	24	5.19
	1–2	356	77.05
	≥3r	82	17.74
BLS-HCP certificate	yes	293	63.41
	no	169	36.58
ACLS-P certificate	yes	13	2.81
	no	449	97.18

OBGY = Obstetric gynecology; PED = Pediatrics; ICU = Incentive care unit; ED = Emergency department; OR = Operation Room; ACLS-P = Advanced cardiovascular life support provider; BLS-HCP = Basic life support healthcare provider; CPR = cardiopulmonary resuscitation.

**Table 2 healthcare-11-00061-t002:** Correlation, Tolerance, and Variation Inflation Factor among the Measured Variables (*N* = 462).

Variables	1	2	3	4	5	6	7	Mean (SD)	Skewness	Kurtosis
	*r*									
	(*p*)									
Demographic variable										
Total career	1								1.07	−0.1
Social variables										
Image	−0.01	1						3.53 (0.72)	0.01	−0.02
	(0.879)									
Subjective norm	0.01	0.27	1					2.71 (0.82)	0.04	−0.1
	(0.928)	(<0.001)								
Organizational variables										
Job autonomy	−0.13	0.29	0.34	1				3.12 (0.59)	0.13	0.51
	(0.006)	(<0.001)	(<0.001)							
Organizational support	−0.05	0.21	0.23	0.45	1			3.19 (0.59)	0.09	0.93
	(0.258)	(<0.001)	(<0.001)	(<0.001)						
PEOU of defibrillators	0.31	0.22	0.18	0.24	0.34	1		3.33 (0.73)	0.45	−0.15
	(<0.001)	(<0.001)	(<0.001)	(<0.001)	(<0.001)					
PU of defibrillators	0.17	0.39	0.03	0.19	0.31	0.4	1	4.02 (0.67)	0.04	−1.01
	(<0.001)	(<0.001)	(0.425)	(<0.001)	(<0.001)	(<0.001)				
Intention to use defibrillators	0.05	0.39	0.24	0.34	0.43	0.59	0.5	3.54 (0.71)	0.29	−0.11
	(0.286)	(<0.001)	<0.001)	<0.001)	(<0.001)	(<0.001)	(<0.001)			
Tolerence limit	0.79	0.75	0.81	0.04	0.11	0.36	0.19			
VIF	1.25	1.33	1.23	1.45	1.43	1.45	1.45			

PEOU = Perceived ease of use; PU = Perceived of usefulness; SD = Standard deviation; VIF = Variance inflation factor.

**Table 3 healthcare-11-00061-t003:** Parameter Estimates for Modified Structural Model and Standardized Direct, Indirect, and Total Effect (*N* = 462).

Endogenous Variables	Direct Effect			Indirect Effect			Total Effect			SMC
Exogenous Variables	*β*	*p*	99% CI	*β*	*p*	99% CI	*β*	*p*	99% CI	
PEOU of defibrillators										
Total career	0.34	0.001	0.26, 0.41				0.34	0.001	0.26, 0.41	
Job autonomy	0.16	0.001	0.06, 0.26				0.16	0.001	0.06, 0.26	
Organizational support	0.28	0.001	0.19, 0.38				0.28	0.001	0.19, 0.38	0.25
PU of defibrillators										
Total career	0.11	0.033	0.01, 0.19	0.09	0.001	0.05, 0.13	0.19	0.001	0.11, 0.27	
Image	0.34	0.001	0.23, 0.43				0.34	0.001	0.23, 0.47	
Subjective norm	−0.15	0.001	−0.24, −0.06				−0.15	0.001	−0.24, −0.06	
Job autonomy	0.01	0.894	−0.08, 0.09	0.04	0.001	0.01, 0.08	0.05	0.294	−0.04, 0.14	
Organizational support	0.18	0.001	0.09, 0.27	0.07	0.001	0.04, 0.12	0.26	0.001	0.17, 0.34	
PEOU of defibrillators	0.26	0.001	0.15, 0.35				0.26	0.001	0.16, 0.35	0.3
Intention of use defibrillators										
Total career	−0.11	0.002	−0.17, −0.03	0.19	0.001	0.14, 0.25	0.08	0.027	0.01, 0.17	
Image	0.14	0.001	0.06, 0.23	0.07	0.001	0.04, 0.12	0.22	0.001	0.14, 0.31	
Subjective norm	0.06	0.081	−0.01, 0.13	−0.03	0.001	−0.06, −0.01	0.02	0.364	−0.03, 0.10	
Job autonomy	0.05	0.182	−0.02, 0.14	0.08	0.004	0.03, 0.14	0.1	0.004	0.04, 0.24	
Organizational support	0.13	0.001	0.05, 0.21	0.18	0.001	0.13, 0.24	0.32	0.001	0.23, 0.41	
PEOU of defibrillators	0.43	0.001	0.34, 0.51	0.06	0.001	0.03, 0.09	0.5	0.001	0.40, 0.57	
PU of defibrillators	0.23	0.001	0.13, 0.31				0.23	0.001	0.13, 0.32	0.51

PEOU = Perceived ease of use; PU = Perceived usefulness; SMC = Squared multiple correlations.

## Data Availability

Not applicable.

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
