# Peer review of "Clinical Nurses’ Intention to Use Defibrillators in South Korea: A Path Analysis"

_healthcare, 2022, doi:10.3390/healthcare11010061_

Round 1
Reviewer 1 Report (Previous Reviewer 2)
The authors made corrections according to my suggestions. The manuscript is written correctly.
I ask the Dear Editor to accept the manuscript for publication.
I accept the manuscript.
Author Response
저희 원고를 검토해 주시고 게재를 수락해 주셔서 진심으로 감사드립니다.
또한 다른 리뷰어의 의견을 받아 수정된 내용을 첨부하였습니다.

Reviewer 2 Report (Previous Reviewer 1)
This study is pretty interesting. But for right understanding of the clinical nurse' intention to use defibrillator, you should have included the factors related to doctors and Korean medical system because doctors and Korean medical system might have a strong influence on clinical nurse's intention to use defibrillator. So, remodeling of study should be needed.
Author Response
"첨부파일을 확인해주세요"

Reviewer 3 Report (New Reviewer)
Dear authors,
The topic of investigation is interesting and has the potential to inform interventions that will improve the use of AEDs within the nursing profession in South Korea.
I have the following comments.
1. The English language throughout the manuscript is poor in quality and makes it difficult to read and understand the concepts. In particular, the introduction and discussion sections are written poorly and need to undergo extensive English editing by a native English-speaking editor.
2. A longitudinal or retrospective cross-sectional study which explored only those nurses who had been exposed to a CPR situation would have been a more appropriate design to test factors that were perceived to have influenced their use/non-use of AEDs. The issue with the cross-sectional design used is that there were a large proportion of nurses for which the perceived barriers were hypothetical in the sense that the nurses had not been exposed to a CPR situation and therefore could only postulate what may be a barrier. This I feel weakens the analysis as it is not strongly associated with what nurses would actually do if placed in this position.
I suggest that if it is statically possible, a further analysis is conducted comparing the responses of those nurses who have actually experienced a CPR situation and did/did not use an AED versus the nurses who have not experienced a CPR situation. This may provide further insight into what are the 'perceived barriers' to using an AED among nurses who have, and have not experienced a CPR situation.
Thank you for your submission and I look forward to seeing an amendment.
Author Response
"첨부파일을 확인해주세요"

Round 2
Reviewer 2 Report (Previous Reviewer 1)
Please attach the questionnaire including Intention to use defibrillators, Perceived usefulness, Perceived ease of use, image, subjective norm, job autonomy and organizational support as supplementary table or figure.
Author Response
Thank you very much for reviewing our paper. We translated the questionnaire into English and modified it into an edited version, and attached the figure
Sincerely,
Gye-hyun Jung.

Reviewer 3 Report (New Reviewer)
Thank you to the authors for implementing the required changes - I wish you good luck with your future research.
I have no further suggestions for amendments.
Author Response
Thanks to your review of our manuscript, we were able to improve the quality of our research.
Our authors are grateful.
Sincerely,
Gye-hyun, Jung.
This manuscript is a resubmission of an earlier submission. The following is a list of the peer review reports and author responses from that submission.
Round 1
Reviewer 1 Report
There is no premise for the extent of the nurse's domain in the Korean medical system. When a cardiac arrest occurs in the hospital, defibrillation requires an order from a doctor, and the implementation can be done by a medical staff (doctor or nurse). Therefore, I think that understanding of the nurse's area should have been dealt with more importantly even in the face of a situation that requires CPR. Without that, this paper, which studies how much other factors influence the implementation of defibrillation, has a very weak problem with the approach. In other words, this study has serious modeling problems, and I cannot admit that this study is an appropriate study.
Author Response
Point 1: There is no premise for the extent of the nurse's domain in the Korean medical system. When a cardiac arrest occurs in the hospital, defibrillation requires an order from a doctor, and the implementation can be done by a medical staff (doctor or nurse). Therefore, I think that understanding of the nurse's area should have been dealt with more importantly even in the face of a situation that requires CPR. Without that, this paper, which studies how much other factors influence the implementation of defibrillation, has a very weak problem with the approach. In other words, this study has serious modeling problems, and I cannot admit that this study is an appropriate study.
Response 1: Thank you very much for reviewing our paper. We added the implementation part as follows, reflecting your advice (from page 10 line 346 to 363)
- Implementation
In IHCA situations, most clinical nurses are the first to be discovered, and their responses can affect the survival rate of cardiac arrest patients [21]. However, skilled clinical nurses are also hesitant to use defibrillators. To solve this problem, first of all, repeated annual BLS training for clinical nurses is also beneficial, but novice nurses must be able to team up with skilled nurses to learn how to use defibrillators and perform IHCAs quickly. Therefore, organizational managers must endeavor to create a positive working environment. Second, educator should develop a curriculum and operate it practically so that nursing students can learn a variety of cardiopulmonary resuscitation techniques from undergraduate courses and increase their ability and confidence. Through these training, clinical nurses can improve their own image. Third, to improve survival rate in IHCA situations, training such as KALS-P (Korean Advanced Life Support)/ACLS-P, as well as BLS, should be carried out continuously and repeatedly to ensure that clinical nurses acquire up-to-date knowledge and skills in CPR. To this end, organizational support, such as changes in organizational policy, will be essential. Finally, support and policy changes from relevant institutions are required to ensure competency development and job autonomy in relation to the use of defibrillators within the permitted work scope of nurses to enhance patient outcomes and clinical nurse professional satisfaction.

Reviewer 2 Report
I had the pleasure of reading and reviewing a manuscript entitled Clinical Nurses' Intention to Use Defibrillators in South Korea: A Path Analysis.
The manuscript stops a very important topic which is defibrillation and its use by medical-nursing personnel. The study conducted is innovative in that it does not present only the amount of defibrillation or its correctness, but focuses on behavioral factors.
The introduction contains the most important information on the presented issue. It is written in a concise manner.
The material and methods of the study are written in a comprehensive manner. The study group was substantial (462).
Statistical analysis was duly performed taking into account the generally accepted norms.
The study received approval from the bioethics committee, which makes it credible.
The tables and figures used are clear.
The discussion is comprehensive.
I suggest:
1. the conclusions be bulleted,
2. add a chapter on the implementation of the obtained conclusions into everyday life, e.g. propose some interesting solutions for using these conclusions.
Once again, I congratulate the authors for their contribution to the study and the elaboration in the form of this manuscript.
Author Response
Point 1, 2: 1. the conclusions be bulleted, add a chapter on the implementation of the obtained conclusions into everyday life, e.g. propose some interesting solutions for using these conclusions.
Response 1: Thank you very much for reviewing our paper. We added the implementation part as follows, reflecting your advice (from page 10 line 346 to 363)
- Implementation
In IHCA situations, most clinical nurses are the first to be discovered, and their responses can affect the survival rate of cardiac arrest patients [21]. However, skilled clinical nurses are also hesitant to use defibrillators. To solve this problem, first of all, repeated annual BLS training for clinical nurses is also beneficial, but novice nurses must be able to team up with skilled nurses to learn how to use defibrillators and perform IHCAs quickly. Therefore, organizational managers must endeavor to create a positive working environment. Second, educator should develop a curriculum and operate it practically so that nursing students can learn a variety of cardiopulmonary resuscitation techniques from undergraduate courses and increase their ability and confidence. Through these training, clinical nurses can improve their own image. Third, to improve survival rate in IHCA situations, training such as KALS-P (Korean Advanced Life Support)/ACLS-P, as well as BLS, should be carried out continuously and repeatedly to ensure that clinical nurses acquire up-to-date knowledge and skills in CPR. To this end, organizational support, such as changes in organizational policy, will be essential. Finally, support and policy changes from relevant institutions are required to ensure competency development and job autonomy in relation to the use of defibrillators within the permitted work scope of nurses to enhance patient outcomes and clinical nurse professional satisfaction.

Reviewer 3 Report
I have read with great interest the manuscript by Dongchoon Uhm and colleagues. The study is clinically useful. Some suggestions might be useful for improving the manuscript.
1. “Materials and Methods” section, how to calculate the sample size? Please report it clearly.
2. The study was based on the analysis of TAM2 questionnaire which was mainly subjective questions. The effect of subjective bias and report bias should be explained clearly.
3. “Results” section, the departments of the nurses should be reported which might influence the use experience of defibrillator.
Author Response
Point 1: “Materials and Methods” section, how to calculate the sample size? Please report it clearly.
Response 1: Thank you very much for reviewing our paper. Following your advice, we have revised as follows (from page 3 line 95 in the Sample and study context to line 99).
This study was conducted to construct the structural model by path analysis with eight observational variables. To use the maximum likelihood estimate, the sample must have at least 200, the more complicated the path, the larger the sample size, so that the tolerance error can be reduced [14, 15]. So, we recruited 500 clinical nurses to achieve a reasonable sample for this study considering the dropout rate. A total of 500 questionnaire packages with cover letters were distributed with a return envelope addressed to the researcher, and 462 questionnaires were analyzed (response rate = 92.4%).
Point 2: The study was based on the analysis of TAM2 questionnaire which was mainly subjective questions. The effect of subjective bias and report bias should be explained clearly.
Response 2: : The contents have been modified as follows (added at page 3 line 116 in the Measurements).
Confirmatory factor analysis was applied to the study variables to assess construct validity. The questionnaire was translated into Korean and back-translated into English to check the validity of the translation, and its content validity was assessed by three experts (one nursing director and two nursing professors). In addition, a pilot test (n=50) was performed prior to conducting the study.
And, added at page11 line 371 in the Limitations to line 375.
Second, this study’s sample was not fully representative of clinical nurses in South Korea, and Third, although the pilot test was conducted to minimize subjective and reporting bias on the self-reported questionnaire used in this study, the social desirability bias and acquiescence responses bias could not be excluded [30]. Therefore, the results of this study might not be generalizable to all clinical nurses in Korea.
Point 3: “Results” section, the departments of the nurses should be reported which might influence the use experience of defibrillator.
Response 3: The contents have been modified as follows (added at page 5 line 179 in the Results and table 1.).
Most participants were female (97.0%). 54.5% of the nurses reported clinical experi-ence of less than five years. Most of the participants’ work departments were general units (Medical/Surgical, OBGY/PED) (62.9%), followed by special units (28.1%).

Reviewer 4 Report
A.- In Population:
¿How did you determine the sample size?
B.- For the Discussion, I recommend to read these papers.
1.- Why is BLS and ACLS a necessity for the modern world´s Cardiac Health.
By INSCOL (Canada) October 25, 2017.
2.- Eman Abd El Aziz Mohamed.
Effect of Cardiopulmonary Resuscitation (CPR) Training Program on knowledge and Practices of Internship Technical Institute of Nursing Students.
IOSR Journal of Nursing and Health Science (IOSR-JNHS) May - June 2017 PP 73-81.
3.- Roberta Kaplow PhD, Sergio Mota DNP.
Nursing roles and responsibilities with Cardio Pulmonar Arrest in Radiology/ Procedural Areas.
Journal of Radiology Nursing 24 June 2022.
4.- Mathilde Staerk, Kasper G. Lauridsen et al.
Barriers and Facilitators for successful AED usage during in-situ simulated in-hospital cardiac arrest.
Resusc Plus 2022. Jun; 10: 100257
5.- Zainah D. Alaryani, Aisha Alhofaian and Mona Elhady.
The relationship between knowledge and self-efficacy of nurses regarding early initiation of cardiopulmonary resuscitation and automated defibrillation in Saudi Arabia.
Belitung Nursing Journal, volume 7, Issue 5, September - October 2021.
C.- In Limitations:
You said, that the strenght of this research, is that It´s the first on this topic in South Korea.
But I found this article, where you are the author:
Dongchoon Uhm PhD, Gyehyung Jung PhD.
"Factors affecting attitudes toward defibrillator use among Clinical Nurses in South Korea: A cross sectional study"
Journal of Emergency Nursing volume 47, Issue 2, March 2021 pages: 305-312.
So, I think your new paper is not the first in this topic in South Korea.
D- E.- These last thoughts (D and E) are only thoughts, not for to correct your research.
D.- It would be very important, that the Ministry of Health and Ministry of Education of South Korea read these papers.
E.- And in the near future, I can read a paper where the attitudes toward defibrillation use, among Clinical Nurses in South Korea are better (after an intervention: training program).
Author Response
Thank you very much for reviewing our paper.
We have revised the contents of the four parts you pointed out, and attached the revised contents in a word file.

Round 2
Reviewer 1 Report
This study has selected the wrong approach to solve problems. The authors should consider the Korean medical system in advance. Then authors mark out appropriate modeling .
Author Response
Response to Reviewer 1 Comments
First point 1: There is no premise for the extent of the nurse's domain in the Korean medical system. When a cardiac arrest occurs in the hospital, defibrillation requires an order from a doctor, and the implementation can be done by a medical staff (doctor or nurse). Therefore, I think that understanding of the nurse's area should have been dealt with more importantly even in the face of a situation that requires CPR. Without that, this paper, which studies how much other factors influence the implementation of defibrillation, has a very weak problem with the approach. In other words, this study has serious modeling problems, and I cannot admit that this study is an appropriate study.
2nd point 1: This study has selected the wrong approach to solve problems. The authors should consider the Korean medical system in advance. Then authors mark out appropriate modeling .
Response: Thank you very much for reviewing our paper. We added the implementation part as follows, reflecting your advice (from page 1 line 26 to page 2 line 54 in Introduction and from page 10 line 362 to 363 in Implementation)
- Introduction
Lately, there has been a lot of interest and development in nurses’ resuscitation per-formance as a cardiopulmonary resuscitation (CPR) team member [1]. Although there is no promise for the extent of nurse’s domain with cardiac arrest (CA) in the Korean medical system, it cannot be doubted that nurses will be performed CPR with defibrillation re-gardless of the time and place of CA.
According to evaluation criteria for healthcare accreditation in South Korea, medical staff (doctors, nurses) and technicians who have direct contact with patients should re-ceive CPR training once every two years [2]. There is no legal problem with clinical nurses use a defibrillator in a situation that defibrillation is required [3]. However, trained clinical nurses are hesitant to use a defibrillator and feeling it as a challenge despite the actual field where required defibrillation. In previous studies [4-6], a lack of knowledge of shockable rhythms and defibrillation, insecurity and anxiety about using defibrillators, recognizing it as a doctor’s assistant job, or fear of negative outcomes were reported as reasons why clinical nurses were not active in using the defibrillator.
The incidence of CA in South Korea is gradually increasing from 21, 905 cases in 2008 (44.3 per 100,000 population) to 30, 539 cases in 2018 (59.5 per 100,000 population) [7]. Considering the rapidly aging society in South Korea, the incidence of CA is likely to increase further. In hospital cardiac arrest (IHCA), patients with specific cardiac diseases, and monitored CA, or wit-nessed CA were related to initial shockable rhythm, and within 3 minutes of patient collapse receives defibrillation, the better their chances of sur-vival [8]. Defibrillation is a powerful tool. But, it is important to know when to use defib-rillation to reset the abnormal rhythm and when not to use it.
The Korean Association of Cardiopulmonary Resuscitation (KACPR) has developed the korean advanced life support-provider (KALS-P) guidelines in 2012 so that healthcare providers working on the front line can undertake professional resuscitation IHCA. The guidelines consist of the two pathways for a CA. There are two shockable rhythms such as ventricular fibrillation and pulseless ventricular tachycardia and two non-shockable rhythms such as asystole and pulseless electrical activity [9]. The Korean Nurses Associa-tion has also recognized and implemented this program as a job training for nurses since 2015. [10]
- Implementation
In IHCA situations, most clinical nurses are the first to be discovered, and their re-sponses can affect the survival rate of cardiac arrest patients [25]. However, skilled clinical nurses are also hesitant to use defibrillators. To solve this problem, first of all, repeated annual BLS training for clinical nurses is also beneficial, but novice nurses must be able to team up with skilled nurses to learn how to use defibrillators and perform IHCAs quickly. In addition, institutions need to consider the installation of AED in each ward for safe and rapid CPR with defibrillation.
